# Properties and Therapeutic Implications of an Enigmatic D477G RPE65 Variant Associated with Autosomal Dominant Retinitis Pigmentosa

**DOI:** 10.3390/genes11121420

**Published:** 2020-11-27

**Authors:** Anna-Sophia Kiang, Paul F. Kenna, Marian M. Humphries, Ema Ozaki, Robert K. Koenekoop, Matthew Campbell, G. Jane Farrar, Pete Humphries

**Affiliations:** 1The School of Genetics & Microbiology, Trinity College Dublin, D02 VF25 Dublin, Ireland; paul.kenna@tcd.ie (P.F.K.); mhumphri@tcd.ie (M.M.H.); campbem2@tcd.ie (M.C.); jane.farrar@tcd.ie (G.J.F.); pete.humphries@tcd.ie (P.H.); 2The Research Foundation, Royal Victoria Eye and Ear Hospital, D02 XK51 Dublin, Ireland; 3Department of Clinical Medicine, School of Medicine, Trinity College Dublin, 2 D02 R590 Dublin, Ireland; ozakie@tcd.ie; 4Trinity College Institute of Neuroscience, Trinity College Dublin, 2 D02 PN40 Dublin, Ireland; 5Departments of Ophthalmology, Human Genetics and Paediatric Surgery, Montreal Children’s Hospital, McGill University Health Centre, McGill University, Montreal, QC H3A 0G4, Canada; robkoenekoop@hotmail.com

**Keywords:** autosomal dominant retinitis pigmentosa, RPE65, incomplete penetrance, 9-cis retinaldehyde, knock-in mouse models

## Abstract

RPE65 isomerase, expressed in the retinal pigmented epithelium (RPE), is an enzymatic component of the retinoid cycle, converting all-trans retinyl ester into 11-cis retinol, and it is essential for vision, because it replenishes the photon capturing 11-cis retinal. To date, almost 200 loss-of-function mutations have been identified within the *RPE65* gene causing inherited retinal dystrophies, most notably Leber congenital amaurosis (LCA) and autosomal recessive retinitis pigmentosa (arRP), which are both severe and early onset disease entities. We previously reported a mutation, D477G, co-segregating with the disease in a late-onset form of autosomal dominant RP (adRP) with choroidal involvement; uniquely, it is the only RPE65 variant to be described with a dominant component. Families or individuals with this variant have been encountered in five countries, and a number of subsequent studies have been reported in which the molecular biological and physiological properties of the variant have been studied in further detail, including observations of possible novel functions in addition to reduced RPE65 enzymatic activity. With regard to the latter, a human phase 1b proof-of-concept study has recently been reported in which aspects of remaining vision were improved for up to one year in four of five patients with advanced disease receiving a single one-week oral dose of 9-cis retinaldehyde, which is the first report showing efficacy and safety of an oral therapy for a dominant form of RP. Here, we review data accrued from published studies investigating molecular mechanisms of this unique variant and include hitherto unpublished material on the clinical spectrum of disease encountered in patients with the D477G variant, which, in many cases bears striking similarities to choroideremia.

## 1. RPE65 Structure, Function, and Localization

RPE65 governs the key enzymatic process of the classical visual (retinoid) cycle in vertebrates [1] whereby vitamin A is converted to visual chromophore, 11-cis retinal, which is required for the formation of light-sensitive retinylidene pigments in photoreceptors, such as rhodopsin in rod cells, which drive phototransduction. Indeed, the speed at which these pigments can be regenerated to provide functioning vision is largely governed by the rate of isomerase activity of RPE65, whereby all-trans retinyl ester is converted into 11-cis retinol, which is then oxidized to 11-cis retinal [2,3,4].

Human RPE65 is a 533 amino acid protein expressed from the *RPE65* gene on Chromosome 1, the gene extending over 21.5 kilobases and containing 14 exons. The protein is iron-dependent and membrane-associated and primarily located in the retinal pigmented epithelium (RPE) where its main function, as mentioned above, is to supply rod and cone opsins with chromophore. RPE65 is also expressed in cone photoreceptors [5], where it may have a role in maintaining homeostasis of retinoid pools rather than in chromophore regeneration [6]. X-ray crystallographic analysis describes a seven-bladed β-propeller surrounding a centrally located iron co-factor and suggests the presence of a retinoid entry/exit tunnel, the entrance of which is membrane bound. The structural analysis also supports the formation of dimers as active enzymatic units, which acquire close membrane contact required for functionality by virtue of specific hydrophobic regions [7,8]. Interestingly, there are other functions of RPE65 including an essential role for proper cone opsin localization and survival [9,10], while recent observations have highlighted a range of secondary attributes including the ability to isomerize lutein to meso-zeaxanthin, which is a constituent of macular pigment along with lutein and zeaxanthin [11], an association with docosahexaenoic acid (DHA), which is an abundant photoreceptor-membrane-associated polyunsaturated fatty acid [12], and possible chaperone activity of retinyl esters associated with retinosomes [13]. On the other hand, as evidenced by the fact that certain patients lack physiologically meaningful functional levels of RPE65 and yet retain some useful daytime vision in their first two decades of life [14] and that *Rpe65* knockout mice (*Rpe*^KO^) respond, albeit minimally, to light stimuli [15], regeneration of the chromophore is not solely the remit of RPE65. There are in fact additional RPE65-independent, light-dependent pathway(s) for 11-cis retinal production that can supply chromophore to photoreceptors especially cones (reviewed by Palczewski and Kiser [16]).

## 2. Mutations within the *RPE65* Gene Causing Autosomal Recessive Inherited Retinal Degenerations (IRDs)

Bi-allelic *RPE65* mutations causing the severe inherited retinal dystrophy, Leber congenital amaurosis (LCA, termed LCA2) or autosomal recessive retinitis pigmentosa (arRP), a somewhat milder disease, albeit with early onset, were first described in the late 1990s [17,18,19,20] and are thought to account for about 7–16% of all LCA cases and 1–2% of those of arRP. LCA2 is characterized by very early-onset night blindness due to the loss of rod photoreceptor function and structure with a concomitant retention of some cones, particularly in the central foveal region, which can provide ambulatory daytime vision into the third decade well after all peripheral vision has deteriorated in some individuals; however, legal blindness by late teens is also a common feature within this patient group [21,22,23]. arRP has a very similar phenotype that differs principally from LCA2 in the lack of nystagmus, the lack of congenital onset, and the later arrival of degenerative and dysfunctional symptoms, with some patients having useful central vision, even in old age [19]. Recently, the number of mutations spread throughout the *RPE65* exome has been reported to be 195, half of these being missense (55%), 21% being insertion/deletions, 13% being splicing defects, 9% being nonsense with indels, and a large deletion and a complex rearrangement comprising the remaining 2% [24].

Research on RPE65 carried out by many groups worldwide in vitro, in rodents, dogs, and in primates facilitated human clinical trials of a gene replacement therapy in which functional copies of *RPE65* were delivered into LCA2 patients’ eyes [25,26,27]. This therapy, voretigene neparvovec, (Luxturna^®^ Spark Therapeutics Inc.), representing perhaps the most significant recent advance in IRD research was approved by the FDA in 2017, the EMA in 2018 and by HC (Health Canada) in 2020, for bi-allelic *RPE65* mutation-associated disease and consists of a disarmed, recombinant adeno-associated virus (AAV)-delivered *RPE65* cDNA construct that is injected subretinally [28]. Indications so far suggest long-lasting substantial gains in visual function including enhanced light perception leading to improved mobility, which continues to be observed four years after therapeutic administration [29]. In fact, a recent report on a 7.5 year follow-up on the initial cohort shows stable visual function after Luxturna^®^ injections [30]. However, in some adult cohorts studied initially, structural degeneration of the retina apparently ensued regardless of functional therapeutic benefit [31], although it is anticipated that this may be overcome by treatment at earlier age and with improved AAV constructs.

## 3. Identification of an *RPE65* Mutation (D477G) Co-Segregating with the Disease in a Family from Ireland with Late Onset adRP

We previously reported the first dominant-acting mutation in *RPE65*, co-segregating with RP associated with choroidal involvement, in two Irish families [32]. Exome capture and next-generation sequencing were used to pinpoint the mutation to c.1430A>G in *RPE65,* which results in a change from aspartic acid (GAA) to glycine (GGT) at amino acid position 477 (p. Asp477Gly or D477G). Mutation to glycine from this aspartic acid residue, highly conserved in vertebrates, replaces a strongly reactive negatively charged amino acid aspartic acid (due to a carboxyl side chain that endows electrochemical behavior to proteins) with a much more stable, non-reactive residue. Indeed, the observed increased electrophoretic mobility of D477G RPE65 compared to the wild type (WT) protein and in silico predicted perturbations [32] reflect the amino acid substitution and suggest that this changed nature of the protein may cause a dominant effect (discussed later). In addition to the prominent choroidal involvement, the retinal degeneration in these families was marked by considerable phenotypic variation, ranging from essentially normal vision to profound visual impairment. Furthermore, given that the proband of the second affected Irish family (distantly related to the original family), a male patient, was initially diagnosed with choroideremia but lacked a mutation in the *CHM* gene, it is possible that the molecular analyses of other such cases may show D477G *RPE65* to be involved. More recently, this same variant and similarly diverse clinical features have been found in several other families in the UK, France, Canada, and the USA [33,34,35] (and Dr Jane Green, personal communication).

## 4. On the Clinical Spectrum of adRP with D477G *RPE65* Mutation

Since 2011, we have been recruiting patients as part of our Target 5000 initiative, which is a collaborative clinical characterization and genotyping study of IRD patients in Ireland involving the Research Foundation at the Royal Victoria Eye and Ear Hospital Dublin, the Mater Misericordiae Hospital Dublin, the Belfast Trust, and the Ocular Genetics Unit at Trinity College Dublin. Currently, 1285 such patients have been assessed at the Research Foundation and recruited for genotyping by a variety of methods including retinal gene panel target capture sequencing, whole gene sequencing, and whole genome analysis [36,37]. As a result of these studies, in total, approximately 50 Irish patients have been identified to date with the D477G *RPE65* mutation.

Patients are examined with particular regard to their best-corrected Snellen visual acuity, Goldmann perimetry, typically using the IV4e, I4e, and 04e targets, Lanthony D15 color vision testing, measurement of the dark-adapted threshold to an 11° white target presented at 15° above fixation (a retinal locus of relatively equal rod and cone photoreceptor density), and Ganzfeld electroretinography (ERG) using the standards advocated by the International Society for Clinical Electrophysiology of Vision (ISCEV) [38], including recording of pure-rod responses, mixed rod and cone responses, and cone-dominated responses to both 0.5 Hz flashes and 30 Hz flickers. Most patients have color fundus photographs taken, as well as spectral domain optical coherence tomography (OCT) to assess the various retinal layers, including the ellipsoid zone (EZ) and the RPE layer. Fundus autofluorescence (FAF), resulting from the fluorescent properties primarily of lipofuscin, is also assessed.

Patients harboring the *c*.1430A>G (p.D477G) *RPE65* mutation report symptoms of impaired night vision in the 4th or 5th decade but, in some, the onset of nyctalopia has been as early as the 2nd decade, while others have no subjective complaints of difficulties with vision in low light even in the 6th decade, which is indicative of some degree of incomplete penetrance (Figure 1). 

Symptoms related to loss of peripheral visual field are likewise variable in age of onset. Intraretinal “bone spicule” or “nummular” pigmentary deposits, varying in density, are evident in most patients as are impaired rod and cone ERG responses. A marked feature in more severely affected individuals is extensive diffuse chorioretinal atrophy, which is reminiscent of fundus changes seen in the X-linked condition, choroideremia.

The spectrum of clinical findings relating to the most mildly affected individual, a moderately affected patient, and one of the most severely affected patients seen at the Royal Victoria Eye and Ear Hospital, Dublin, in whom the *RPE65* D477G mutation was identified, are summarized in Table 1. Figure 2 illustrates the marked difference in phenotype between the mildest and most severely affected individuals.

Most patients found to harbor this mutation have features that are intermediate between these two extreme poles, as exemplified by the moderately affected patient in Figure 3.

In common with observations of Hull et al. [33] and Jauregui et al. [34,35], FAF imaging shows outer retinal whorls or possibly tubulation and chorioretinal atrophy indicated by hypofluorescent patches (Figure 3A). However, in contrast to the former study, there appears to be some evidence of lipofuscin accumulation in the central retina where the RPE remains intact. This is an interesting finding, given that biallelic RPE65 patients generally show no such deposits due to the lack of isomerase activity and thus a lack of all-trans retinal, from which A2E, a major constituent of lipofuscin, is formed [23]. Generally, the phenotype associated with this IRD is compatible with relatively good visual function and is slowly progressive. Most patients retain useful vision, even in later life, in marked contrast to the severe phenotype generally associated with bi-allelic *RPE65* mutations.

## 5. Modeling the Molecular Pathology of D477G RPE65 in Mice

### 5.1. Introduction

In order to probe further into the etiology of the D477G variant, in vitro studies conducted using transient expression of cloned human *RPE65* cDNAs in mammalian cell cultures were reported in Kenna et al. [39]. Isomerase activity of the mutant enzyme in the presence of substrate was found to be approximately 75% that of WT, which is a result that is slightly higher but consistent with the reduced level observed by Li et al. [40] under their minimal 11-cis retinal assay cell culture conditions. More importantly, the expression of a 1:1 combination of D477G and WT proteins (the same total quantity as used in the single construct transfections) produced an intermediate level of chromophore in both studies, indicating that, at least in a minimal isomerase system in vitro, enzymatic activity is not impacted negatively by D477G variant enzyme. That is, we do not observe a dominant negative effect of the mutant isomerase over wild-type function. Further supporting these data, Li et al. [40] also showed that other D477 amino acid substitutions had little effect on 11-cis retinal production and noted that from an evolutionary perspective, glycine rather than aspartic acid most frequently occupies the 477 paralogous position in ancestrally related proteins. Given these initial results, we then looked to see if the D477G RPE65 protein per se could have an effect on general cell metabolism in ARPE-19 (an immortalized human adult RPE cell line) cultures and used the MTS tetrazolium-based colorimetric assay [41] to measure outcome. Comparison of cultures grown under normal conditions and transfected with WT *RPE65*, D477G *RPE65*, or both cDNA constructs yielded no differences in MTS readout, which is an indicator of cell viability. However, mild stressing of the cells by seeding at a low density prior to transfection enabled a small but significant deficit in cellular metabolism to be detected in cultures expressing D477G RPE65 protein, which was also observed in cells transfected with both cDNA constructs [39]. Taken together, these findings suggest that the D477G variant protein may have a dominant negative or gain of function toxic effect over the WT RPE65, which affects normal cellular metabolism and has nothing to do with isomerase activity.

Given that this dominantly inherited condition causes a form of retinitis pigmentosa affecting the RPE, choroid, and retina in humans with a cone-rich macula that is absent in mice, mouse models can never be expected to fully recapitulate patients’ phenotypes. Nevertheless, animal models have provided valuable mechanistic insights into other forms of RP [42,43,44,45,46] in addition to being available as vehicles in which to test emerging therapies based on such insights [47,48,49]. For these reasons, four laboratories, including our own, deemed it worthwhile to create mouse models of D477G RPE65-associated adRP.

### 5.2. AAV-Mediated Expression of Human RPE65 in Wild-Type Mice and Knock-In (KI) Model

Our first model reported in Kenna et al. [39] was created by expressing human *RPE*65 promoter-driven D477G *RPE65* cDNA in WT C57Bl6/J mouse eyes following subretinal delivery via AAV using a similar methodology as that used previously to model RP10 [50]. Wild-type mice were injected at PND 31 with right eyes receiving D477G *RPE65* construct and left eyes receiving the WT equivalent (Figure 4A). Then, 55 days later, there was a 42% and 33% difference in composite maximum mixed rod and cone responses from right eyes compared to contralateral eyes for a- and b-waves, respectively [39]. Retinas from two injected animals that exhibited different degrees of ERG deficit (Figure 4B) were examined histologically. This analysis was performed blind on sections from equivalent positions with respect to optic nerve head and superior/inferior and temporal/nasal locations. No difference in the outer nuclear layer (ONL) thickness between contralateral eyes of each animal was observed (Figure 4C), suggesting that, at least in these two animals, there may be a disconnect between photoreceptor dysfunction detected by electrophysiology and mechanisms leading to rod and cone cell death. Taken together, these data demonstrate that the exogenous expression of human D477G RPE65 but not WT RPE65 protein in adult wild-type mouse eyes reduces functional output, which suggests that contrary to implications of in vitro data [39], a dominant negative influence of human variant protein over normal endogenous protein as regards enzymatic activity may exist. On the other hand, based on preliminary ONL thickness assessment, there is no evidence for any negative structural effect, which is an unexpected finding given the obvious degeneration observed in patients. This could be due to the absence of the mutant protein during development of the mice or that insufficient time had elapsed between the delivery of AAV and animal sacrifice (55 d) for structural deterioration to be detected. In addition, should expression in cones of the variant protein be involved in photoreceptor degeneration, the paucity of transgene expression in cones due to the use of AAV serotype 9, which preferentially targets RPE rather than photoreceptors in mice [51], may have obscured observation of such an outcome. These data, although interesting, are further complicated by high expression levels of AAV-delivered cDNAs, in addition to normal quantities of the endogenous WT murine RPE65 protein, which may have exaggerated any effect of the aspartic acid to glycine substitution.

An initial D477G *Rpe65* KI mouse model constructed using crispr/cas9 technology (kind gift from Dr. Thomas Ferguson, Washington University, USA) was found to have no discerning structural or functional phenotype, as assessed by ERG and retinal histology respectively, when compared to wild-type littermates, in either heterozygous or homozygous states. For these reasons, a more elaborate D477G *Rpe65* KI mouse was created by homologous recombination in which two further changes, M450L and Q475H were also introduced (Figure 5A). The reasoning behind the former substitution was twofold: to see whether the associated higher rate of isomerase activity [52] would force an exaggerated phenotype and thus facilitate experimental studies, and secondly, along with Q475H, to “humanize” the expressed protein in regions near the principal mutation. However, despite many different examinations in both heterozygous and homozygous mice, including looking for delayed dark adaptation and thinning of the ONL, structural, or functional phenotypes of note were again not observed (Figure 5B).

Three further D477G *Rpe65* KI mice models constructed independently by other research groups [40,53,54] are all very similar but with subtle differences due to methods of creation and preferences with regard to the number and position of “humanizing” amino acid residue changes (Table 2). Taken as a whole, the structure and functioning of visual systems in the various models are similar, with minimal phenotypic deficits observed. However, research directions taken by each laboratory have been somewhat different and have thus shed light on several diverse aspects of possible disease mechanisms that could translate into future therapeutic targets (Table 2).

### 5.3. Possible Dominant Negative Effect Observed in KI Model

The KI model by Shin et al. [53] was constructed using site-specific homologous recombination and resulted in a mutated allele that was identical to the endogenous wild-type counterpart except for the c.1430A>G substitution resulting in D477G and a 101bp insertion into the intron between exons 13 and 14. In contrast to ocular findings in patients, there were no obvious structural or functional deficits in the animals’ retinas except for the vastly reduced abundance of RPE65 protein within the RPE of homozygous mice (Table 2). However, on close histological examination by TEM, lipid droplets were observed in the RPE of a 13-month-old heterozygous mice, which was similar in appearance to those described by Redmond et al. [15] in *Rpe65*^KO^ mice and ascribed to retinyl ester accumulation. In addition, delayed dark adaptation, a feature of human D477G *RPE65* pathophysiology, was investigated in rods and cones following either mild or strong photo-bleaching, respectively. Rod photoreceptor responses elicited by heterozygous mice were observed to be normal, while those from cones showed a delay. Moreover, the authors went on to demonstrate that this deficit corresponded with reductions in the synthesis of 11-cis retinal and the consumption of all-trans retinyl ester in heterozygotes, indicating an overall reduction in the isomerase activity of approximately 50% compared to WT mice (Table 2). Interestingly, retinoid levels of WT/*Rpe65*^KO^ heterozygotes that underwent the same protocol were also examined and found to be the same as those observed in WT mice and significantly different to the KI heterozygotes. Thus, the authors concluded that D477G acts in a dominant negative manner over WT as regards isomerase activity and that this affects dark adaptation in a manner analogous to that observed clinically. However, it is hard to reconcile a level of 50% of WT isomerase activity and the corresponding reduction in amount of available chromophore observed in KI heterozygotes, with severe pathology seen in patients, unless this possible dominant negative property of D477G is somehow much greater in humans than it is in mice. This may in fact be true if the higher rate of isomerase activity required for normal vision in humans is more susceptible to disruption by the variant than the relatively slower rate observed in mice. In addition, it would be interesting to determine whether the observed dominant negative effect actually relates to a reduction by D477G on WT enzyme activity or whether an unrelated but potentially toxic gain of function effect of D477G protein could possibly compromise total cellular function with knock-on effects, which could include isomerase activity.

### 5.4. A Dominant Phenotype Observed in KI Model and D477G RPE65 Protein Modeling and Crystallography Suggests Novel Properties

Choi et al. [54] produced a very similar KI model to that of Shin et al. [53], except for a smaller 85 bp intron insertion. Both of the models have methionine at position 450 of the RPE65 protein sequence, and thus, the slower isomerase kinetics associated with C57Bl6/J wild-type mice, which were used as controls and for breeding purposes. Protein expression was again reduced to approximately 50% and 20% of WT values for heterozygotes and homozygotes, respectively (estimated from Western blot band intensities), and the latter genotype also exhibited enhanced ubiquitination and degradation of RPE65 protein levels compared to the former, while little or no degradation was associated with WT mice. In contrast to the findings of Shin et al. [53], small but significant structural and functional anomalies were described in heterozygous mice that were similar or slightly more pronounced in homozygotes. Therefore, at 9 months in both genotypes, ONL measurements were reduced by about 10% of WT thickness, while significant deficits were also observed in both scotopic and photopic ERG recordings (Table 2). Indeed, the authors even report a similar number of fundus autofluorescent spots, which were thought to be due to inflammation, at 9 months in both heterozygotes and homozygotes (although they were also observed at earlier timepoints in homozygotes). This apparent near-equivalence of phenotypes regardless of whether a single or both alleles carrying the D477G *Rpe65* mutation are present, emphasizes the dominant nature of this mutation and is in stark contrast with the reported retinoid profiles, which suggest that regeneration of 11-cis retinal or levels of retinyl esters following photo-bleaching follow a statistically significant step-wise increase or decrease respectively from WT through heterozygote to homozygote. Looking at these data, it is challenging to envisage that such a clear (albeit subtle) dominant structural and functional phenotype could stem from a semi-dominant biochemical observation in which, according to the literature, the heterozygote should have ample chromophore for normal vision [55]. Then, the authors turned their attention to possible novel physicochemical properties of human D477G RPE65 protein. The fact that Asp477 is a typical “gatekeeper” residue, i.e., provides hydrophilic properties to a hydrophobic region of a protein, and Gly477 is not, suggests that D477G mutation could increase hydrophobicity and thus endow the protein with a propensity to form novel interactions with other cellular constituents. Then, proof of such a theory was provided by these researchers through in silico protein modelling and X-ray crystallography. They first demonstrated that a model consisting of a 9-residue peptide representing the region around D477 fused to the appropriate region of the related (and more easily crystallized) carotenoid cleavage enzyme (ACO) protein to form D477-ACO is distorted by mutation to G477. Crystals made from these two protein chimaeras that differed by just the aspartic acid to glycine substitution were fundamentally different in nature, and careful examination of the crystals’ molecular packing highlighted novel protein–protein interactions in the Gly477-ACO chimaeric proteins, which were absent in the Asp477 equivalent. These interactions were predominantly hydrophobic and while not predicted to change, the tertiary structure or active site of RPE65, could, as pointed out by the authors, encourage aberrant, novel arrangements and contacts of the protein with other cellular components. Thus, it is not too difficult to imagine that a new interaction based on the changed physicochemical properties of D477G RPE65 could result in a neomorphic, toxic gain-of-function mutation whereby a single gene dosage as in heterozygous KI mouse or human patient results in much the same phenotype as the bi-allelic mutant homozygote. In this regard, it should be noted that RPE65 has been proposed to be involved in processes that would not be predicted to require isomerase activity, such as cone opsin localization [9] storage or the transport of retinyl esters associated with retinosomes [13] control of retinoid levels in cones [6] and DHA binding [12], which may be perturbed by abnormal contacts.

### 5.5. Chronic Light Exposure Endows KI with Subtle Mutant Phenotype; Aberrant Splicing a Possible Disease Mechanism

A fifth D477G KI murine model on the C57Bl6/J background (thus also Met450) was constructed using crispr-cas9 technology by Li et al. [40], and thus, it differs from those of Shin et al. [53] and Choi et al. [54] by the absence of additional sequences within the intron following exon 13. The mutant allele also differs in being engineered to resemble more closely the human diseased counterpart. Specifically, the following changes were made in addition to c.1430A>G: c.1425A>C resulting in Q475H substitution together with silent changes c.1434T>C and c.1435C>T. Strikingly, in 7-month-old animals, the RPE65 protein level in homozygotes was even lower than in the other models at 3% that of WT, while that of heterozygotes was similar to previous reports at about 50%. This begs the question as to whether extra changes to humanize the KI gene result in reduced expression or increased degradation of this engineered allele compared to those of Shin [53] and Choi [54]. On the other hand, does humanizing the KI gene and leaving intron 13 unchanged result in a truer expression of what actually occurs in humans? In general, with regard to the model from Li et al., [40], observed ERG deficits were less pronounced than those reported in the other two models, and there was a degree of cone opsin (OPN1SW) mis-localization noted. There were no differences in ERGs between WT and either mutant genotype after standard overnight dark adaptation, even at 11 months, although retinyl ester levels were significantly increased in mutant animal eyes with 11-cis retinal levels remaining unaffected. However, impaired rod responses were detected in homozygous mutants if the dark adaptation period was shortened to 20 min, and in this case, not only was retinyl ester accumulation evident but also greatly reduced levels of 11-cis retinal. Most importantly, chronic low dose light exposure (500 lux) for 6 months reduced cone ERG recovery time, decreased ONL thickness by approximately 13%, and led to the presence of vacuoles in RPE in KI/KI mice (Table 2). In contrast, there were no clear structural or functional deficits observed in heterozygous animals, suggesting a lack of a dominant negative effect. However, the main findings reported in this paper by the Redmond laboratory concern the discovery that the c.1430A>G mutation results in defective splicing in both the murine model and in a cell culture system in which a human D477G *RPE65* minigene is expressed. Only a small minority of transcripts from mutant alleles are reportedly translated into bona fide D477G proteins; these presumably provide enough isomerase activity to sustain vision in homozygous KIs under normal vivarium conditions. The vast majority of the variant transcripts are thought to be eliminated through nonsense-mediated decay. However, the authors go on to infer that aberrant splicing may be principally responsible for the pathogenicity of the variant, because this phenomenon results in vastly reduced levels of D477G RPE65 protein, thus making isomerase activity rate-limiting with age or increased stressors, especially light exposure. Approximately one-third of all possible variant transcripts were studied, and the results showed that some appear to have the potential to generate protein products. The authors tested a single such variant for isomerase activity, which was found to be lacking due to the rapid degradation of the mutant protein. Splicing defects resulting in low levels of chromophore are clearly detrimental to vision in homozygous animals; however, carriers of null *RPE65* mutations are generally reported to have normal vision due to the presence of a functioning allele [18,19]. Interestingly, a notable exception to this, reported by Felius [56], involves a carrier of a splice site mutation who, unlike his carrier father, did exhibit mild visual deficits, suggesting, at least, that in cases of mutation within intervening sequences, there may be some variability in expression in carriers. Theoretically speaking, an alternatively spliced transcript could exist that when translated into protein could have an extremely strong dominant negative effect, thus rendering the WT enzyme inactive—or it could, as pointed out by the Li et al. [40], adopt a toxic gain of function and negatively impact retinal health in man. However, this was clearly not the case in this animal model given the mild phenotype observed in heterozygous mice.

### 5.6. Modeling the Molecular Pathology of D477G RPE65 in Mice—Conclusion

In summary, research into the molecular pathology of D477G RPE65 has revealed that this variant leads to a very mild phenotype when modeled in mice, which may only in part be caused by the reported moderate reduction in isomerase activity. These observations are not surprising, given that the more severe clinical phenotypes seen in biallelic RPE65-IRDs are associated with greatly reduced or non-detectable RPE65 activity and are generally absent in carriers [17,18,19,20,21,22,23,24]. However, subtle deficits in rod and cone function [39,40,53,54], minor cone opsin mis-localization [40], and the presence of vacuoles, lipid droplets, and autofluorescent spots in outer segments [40], RPE [53], and fundus [54] respectively, are suggestive of pathological mechanisms, which although a minor feature in mouse models, could be more exaggerated in patients. Furthermore, it is notable, as evidenced by a comparison of the various knock-in models, that even within the inbred C57Bl6/J mouse strain, the D477G RPE65 variant can give rise to (subtly) different phenotypes in a manner similar in nature (but not in magnitude) to the clinical spectrum observed in patients, which ranges from asymptomatic through “classical RP” to severe macular atrophy and choroidal degeneration [32,33,34,35,39] and Section 4, above. Future studies using patient-derived cell lines and retinal organoid cultures are urgently needed to investigate further the various hypotheses highlighted by animal studies.

## 6. Oral Synthetic Retinoid Therapy–Animal Studies Lead to Clinical Trials

As indicated by the success of voretigene neoparovec gene therapy, as outlined in Section 2 of this review, the supplementation of 11-cis retinal should also, in principle, restore visual function in those forms of hereditary retinal degeneration caused by mutations in genes encoding enzymatic components of the retinoid cycle in which the re-cycling of all-trans retinal into the 11-cis form is compromised. However, 11-cis retinal is highly unstable and not suitable for development as a pharmacological agent. On the other hand, synthetic 9-cis retinyl acetate, which is converted in vivo into 9-cis retinal, is very much more stable and capable of forming iso-rhodopsin and initiating the visual transduction cycle. Early studies showed that *Rpe65*^KO^ mice, which lack 11-cis retinal and thus a functioning visual cycle, were able to elicit visual responses when treated with 9-cis retinal, which is an easily synthesized, thermodynamically stable analogue of the natural chromophore [57]. Light sensitivity, due to the formation of iso-rhodopsin, which can function as a replacement to rhodopsin in phototransduction, was proportional to retinoid dose and was sustained long-term in animals receiving treatment at a young age. In addition, retinyl ester deposits visualized as oil droplets within RPE cells and abundant in untreated eyes were greatly reduced in treated *Rpe65*^KO^ mice [58], while Batten et al. [59] reported increased rod outer segment lengths in a similar model analyzed directly following a multiple gavage regime. This finding was confirmed by Maeda et al. [60] and extended to show that degeneration of the ONL was retarded in animals dosed weekly for 6 months. This study also demonstrated that the treatment of mice receiving an equivalent of 50-fold the dose used in clinical trials was both efficacious and well tolerated. An intravitreal delivery of 9-cis retinal also improved vision in the Briard dog model [61], the latter carrying a four base-pair recessive deletion within the *RPE65* gene. The development of stable prodrugs based on 9-cis retinyl acetates, which generate 9-cis retinal, enabled oral dosing to replace gavage in animal studies and led to the generation of the drug QLT091001 (Retinagenix Therapeutics), which has shown promise in clinical trials for LCA and arRP [62,63].

Regarding assessments of the potential efficacy of 9-cis retinal therapy in human subjects, three investigations having so far been reported: two of these in patients with LCA [62] and arRP [63] caused by bi-allelic recessive mutation within the *LRAT* or *RPE65* genes and one in subjects with late onset adRP expressing the heterozygous, dominant RPE65 variant D477G [39], the latter suggesting that this treatment may represent a therapeutic option for dominant D477G *RPE65* patients. Likewise, voretigene neparvovec may also be a therapeutic option for these patients; indeed, QLT091001 may prove a useful combination therapy with the gene medicine. Furthermore, 9-cis retinal is a low molecular weight compound (284.4 Da), and while access to the retina following systemic administration would be expected to be hindered by the inner blood–retina barrier, access into the RPE (where the molecular pathology manifests) from the underlying choroidal circulation is possible. In each of these open label phase 1b clinical studies (all lacking a placebo group), patients were orally administered 9-cis retinaldehyde acetate (10–40 mg/m^2^ per day) for 7 days, and its effects on visual acuity (VA) and visual function (VF) were monitored for periods of up to 12 months following this 7-day treatment regimen. In the study reported by Koenekoop et al. [62], of the fourteen patients aged 6 to 38 years old (average age 18) and enrolled, 10 showed an improvement in VF area of 28–683%, while six showed VA improvement of 2–30 letters. The effect of the single one-week regimen was sustained over long periods, the median duration of response being approximately 160 days. fMRI scans showing blood oxygen improvements, which were correlated with visual function in the four patients that were analyzed.

Scholl et al, [63] subsequently assessed the effects of this therapy using the same regime in 18 older arRP patients (average age 28.5 years old). In that study, increases in functional retinal area were observed in 12 patients, the mean duration of the response being approximately 11 weeks, while all but one patient showed increases in VA within 2 months of treatment, these responses lasting up to 4 months. Interestingly, the greater photoreceptor outer segment length (EZ length) at study baseline, as measured by OCT, was positively correlated with treatment efficacy.

The rationale adopted by Kenna et al. [39], in assessing potential effects of oral retinoid therapy in members of the large family from Ireland in which the heterozygous D477G variant was initially identified, is that a significant component of the pathology associated with the condition may involve reduced RPE65 enzymatic activity; hence, augmenting levels of chromophore might possibly render remaining photoreceptor cells functional for longer periods of time. Five subjects were enrolled in this Phase 1b proof of concept (PoC) study. Three of these five aged 67–68 years achieved VF improvements of between 70 and 200% of baseline values with maximal responses at 7–10 months. An example from a typical response from one patient is given in Figure 6. Of two VF non-responding patients, one improved and maintained VA from 5 to 15 letters for six months post-treatment. Two aspects of this PoC study are worthy of emphasis. In studies of retinoid therapy in LCA and recessive RP patients [62,63], improvements in vision began relatively early and were sustained over a shorter period than in the study reported by Kenna et al. [37], where visual improvement commenced later but was recordable for at least 10 months post-treatment. It is also encouraging to highlight the differences in the age of patients in these studies. The average age of bi-allelic *RPE65* LCA and arRP patients was 18 and 28.5 respectively given the early onset severe nature of these diseases [62,63], whereas the D477G *RPE65* patients, while all also having advanced disease, were on average 62 years old [39]. Quoting Kenna et al. “It is remarkable that after 62 years of retinal disease, it is possible to reactivate dormant photoreceptors with a simple retinoid” [39]. Therefore, this may hold promise for younger patients with early stage disease, where periodic retinoid therapy could prove beneficial over prolonged periods of time. This Phase 1 proof-of-concept study evaluated the safety and efficacy of the synthetic retinoid. The therapy was delivered orally, and thus, it was not possible to do a placebo-controlled trial or to compare the results in treated versus untreated eyes. As a proof-of-concept study, the small number of patients treated did not permit the generation of data that could be subjected to statistical analysis. There were no serious adverse clinical reactions reported in any of these studies.

## 7. Conclusions and Prospects

While it is clear that the supplementation of chromophore in D477G *RPE65* patients can help to restore vision, the prevention of further retinal degeneration in this cohort remains a challenge. However, research largely carried out in KI murine models has revealed several interesting phenomena associated with the variant, including alternative splicing, physicochemical changes, possible negative dominance, with observations of oil droplets, vacuoles, autofluorescent bodies, and some cone opsin mislocalization [40,53,54], any one, or all of which, may underlie the structural destruction of RPE, choroid, and retina observed in humans. In common with recessive *RPE65*-associated IRDs, the accumulation of excess non-recycled retinoids due to a lack of isomerase activity has been suggested as a possible contributing factor in disease pathology [15,23]. However, in this scenario, two mutant genes are present, since carriers of these null mutations who also express a normal copy of *RPE65* gene generally do not manifest symptoms [18,19]. Given that D477G *RPE65* patients should also, in theory, have a functioning *RPE65* copy, it is difficult not to conclude that there must be some level of dominance of D477G RPE65 over the WT protein. Such a dominant effect could arise from aberrant splicing resulting in a novel variation of D477G RPE65 [40] or from a changed binding property of the protein [54] in at least two ways. Firstly, interaction of the novel/altered protein with WT RPE65 could greatly reduce enzymatic function in humans. However, this does not seem to be the case in mice, because heterozygous KIs were either found to have approximately 50% the activity of WT [40,54] or, when compared to WT/KO heterozygotes, only a mild dominant effect over isomerase activity, which again reduced levels of 11-cis retinal to 50% WT values [53], which is a level in line with normal visual functioning in most *RPE65*-associated IRDs [18,19]. In a second scenario, given the observation that a single copy of D477G *Rpe65* can result in structural deficits in mice and that the reduced ONL thickness does not appear to be significantly worse in homozygous mice carrying two copies [54], it is possible that D477G RPE65 manifests as a toxic gain of function protein.

It would be of great interest to see whether such a novel D477G RPE65 form is found within products of alternatively spliced transcripts because, given the fact that the relative quantities of these products are likely to vary even between closely related individuals and also with age, this may lead to an understanding behind the incomplete penetrance and the variable timing of disease onset reported in Bowne [32]. However, the mechanism of such hypothetical toxicity may more readily be explained by illegitimate associations of D477G RPE65 protein with other cellular components as discussed by Choi et al. [54], or indeed, the absence of an essential binding event due to the variant. Extending these research avenues into the future would also be greatly facilitated by the creation of patient-derived cell- and organoid-based models similar to those developed for LCA2 and RP2 [64,65]. In this regard, Ding et al., [66] have produced induced pluripotent stem cells (iPSC) lines derived from three affected family members from the original publication of Bowne et al. [32], in addition to age-matched controls. The differentiation of these iPSC lines into RPE cell lines and retinal organoids in combination with genome editing strategies would undoubtedly provide superior models of this form of adRP and could also provide proof of principle for in vivo gene editing therapy or form the basis for personalized regenerative medicine. In the meantime, treating younger D477G *RPE65* patients with retinoid therapy may not only improve vision but could also retard retinal degeneration by reducing photoreceptor damage caused by the continued activity of non-liganded opsin apoproteins [67].

## Figures and Tables

**Figure 1 genes-11-01420-f001:**
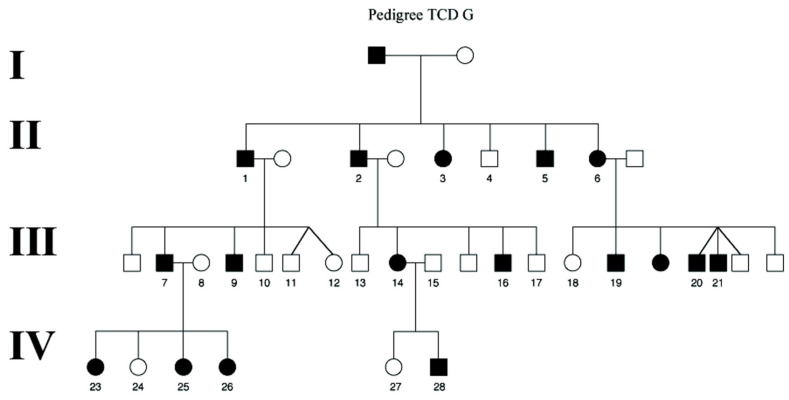
TCD G, the largest Irish pedigree in which the *RPE65* Asp477Gly mutation segregates. Black filled symbols indicate affected individuals carrying the D477G *RPE65* mutation and also the asymptomatic individual III-9. Unaffected females are designated by unfilled circles and males by unfilled squares.

**Figure 2 genes-11-01420-f002:**
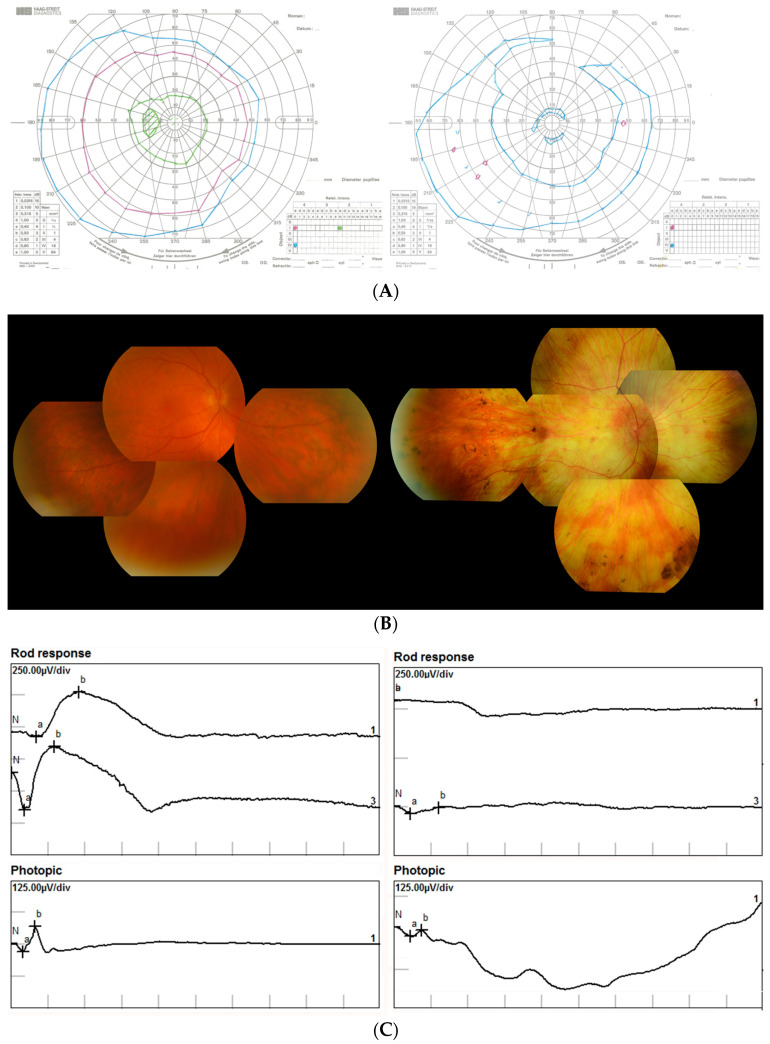
Left panels: clinical features of asymptomatic individual, aged 64 years. Right panels: Clinical features of severely affected individual, aged 55 years. (**A**) Goldmann Visual Fields (Right eye). Left panel shows normal fields to all three targets Blue IV4e, Purple I4e and Green I2e; Right panel show extensive central and mid-peripheral field loss, even to the largest target (blue). (**B**) Fundus images (Right eye). Left panel shows normal fundus appearance; Right panel shows extensive changes with marked choroidal thinning. (**C**) Electroretinography (Right eye). Left panels show normal rod-dominated responses and slightly reduced cone-dominated responses (Photopic). Right panel shows very significantly attenuated rod-dominated and cone-dominated responses. N = initiation of flash stimulus, a = peak of 1st negative deflection, b = peak of subsequent positive deflection. (**D**) Optical coherence tomography (Right eye). Left panel shows normal retinal structure. Right panel shows extensive retinal thinning, with loss of photoreceptors.

**Figure 3 genes-11-01420-f003:**
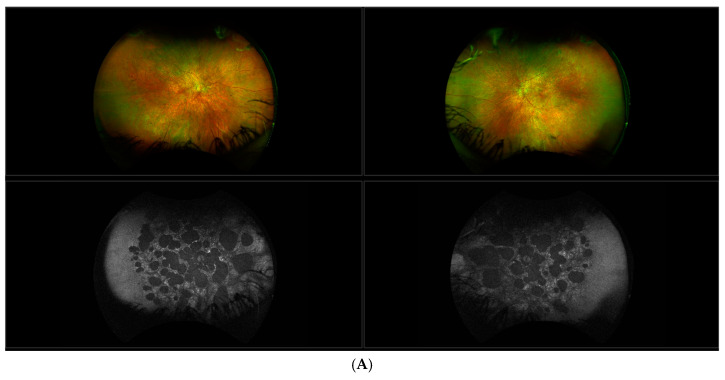
Clinical features of a 47-year-old moderately affected female. (**A**) Fundus photographs (top) and Fundus autofluorescence (FAF) images (bottom). Note the patchy areas of hypofluorescence in FAF images, which are indicative of loss of retinal pigmented epithelium (RPE) and overlying retina. Spots of hyperfluorescence, suggestive of increased accumulation of lipofuscin within RPE cells, are evident at the margins of the hypofluorescent areas. The hypofluoresence is mirrored in the multiple scotomas (**B**) delineated on Goldmann perimetry (Green = III4e target, Orange = II4e target and Red = I4e target) and in the patchy area of retinal thinning evident in the OCT of the left eye adjacent to a whorl-like disturbance in the photoreceptor layer (right panel (**C**)).

**Figure 4 genes-11-01420-f004:**
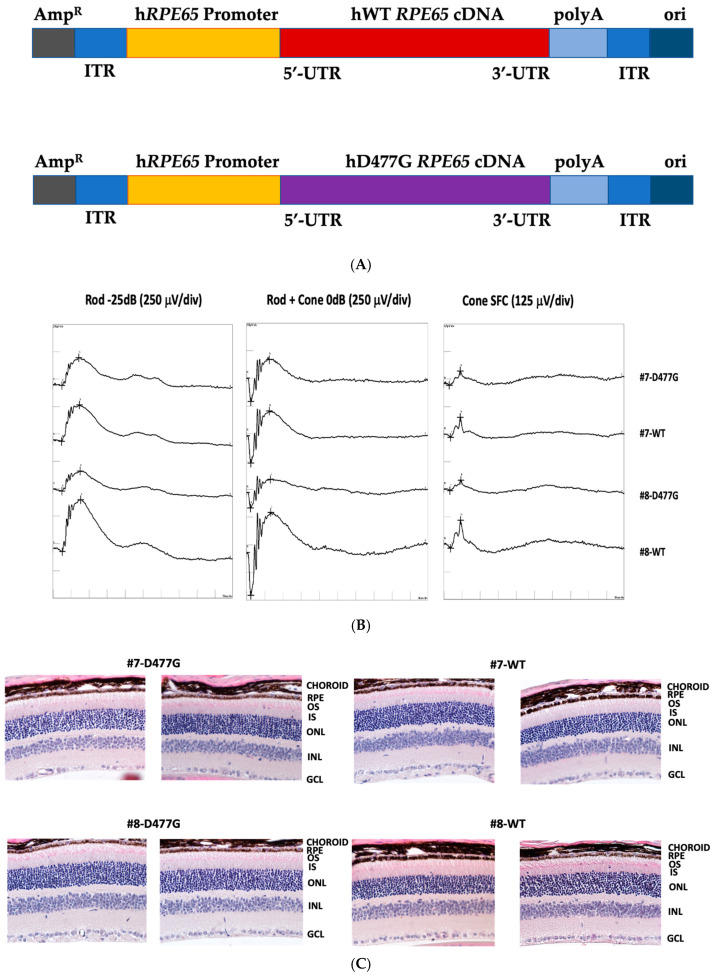
AAV9-mediated expression of human *RPE65* in mouse eyes. (**A**) Construct design for the expression of human WT *RPE65* (top) and D477G *RPE65* (bottom) cDNAs. (**B**) Examples of reduced ERG amplitudes observed in D477G *RPE65*-expressing eyes compared to contralateral WT *RPE65*-expressing eyes in mice #7 and #8. (**C**) No difference observed in gross retinal histology between contralateral eyes of mice #7 and #8 expressing D477G *RPE65* or WT *RPE65* cDNAs.

**Figure 5 genes-11-01420-f005:**
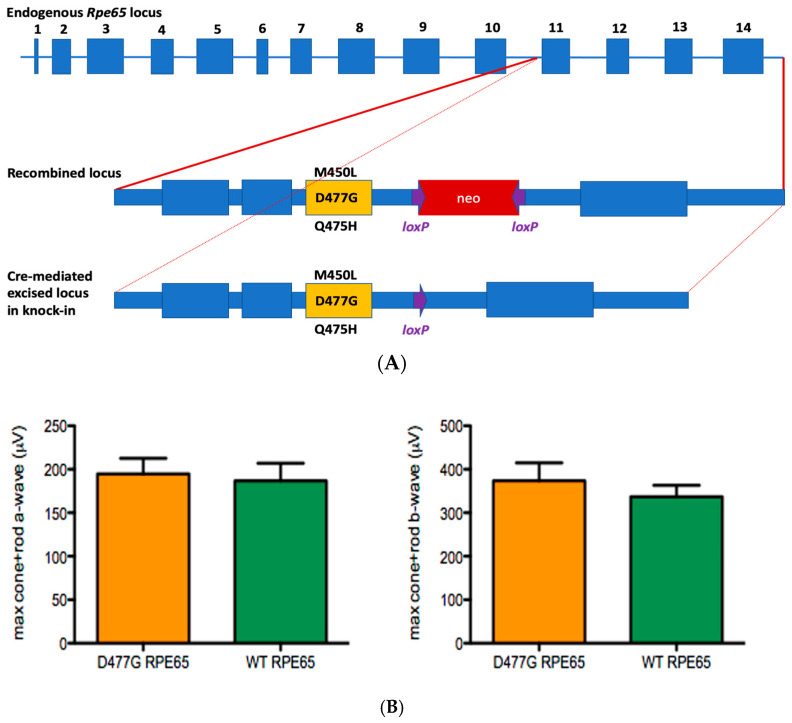
Studies in a D477G, M450L, and Q475H *Rpe65* triple knock-in mouse model. (**A**) Generation of the model by homologous recombination. (**B**) No differences in ERG amplitudes were observed between homozygous D477G *Rpe65* triple knock-in model and wild-type littermates (*n* = 5).

**Figure 6 genes-11-01420-f006:**
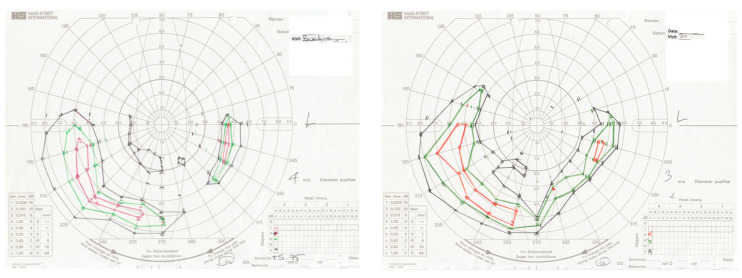
Improvement in visual fields following treatment. Goldmann kinetic visual fields (left eye) of participant 010802 showing expansion of the visual field area following 7 days of oral treatment with QLT091001 at 40 mg per m2. Left panel: baseline, pre-treatment visual field. Right panel: 30 days post-treatment visual field. Target color codes: V4e = black; III4e = green; II4e = orange; I4e = red.

**Table 1 genes-11-01420-t001:** Summary of clinical findings in an asymptomatic, a moderately affected, and a severely affected patient.

	Asymptomatic	Moderately Affected	Severely Affected
Age at Exam	64 years	47 years	55 years
Night Blindness	No	Yes	Since age 20 years
Visual Field Loss	No	Yes	Since age 40 years
Best-corrected Visual Acuity	6/6 (normal) in right and left eye	6/6 in right 6/6_−1_ in left	6/60 (Right); 6/19 (left)
Driving	Yes	Yes	Stopped in early 40 s
Cataract	No	No	Yes (not visually significant)

**Table 2 genes-11-01420-t002:** Summary and comparison of data generated from six mouse models of D477G *Rpe65*.

D477G/*RPE65* Mouse Model	Construction Method	Changes Additional to D477G (c.1430A>G)	Amino Acid at Position 450	% WT Protein Expression	% WT All-Trans Retinyl Ester Level ^1^	Phenotype of WT/KI ^2^	Phenotype of KI/KI ^3^	Additional Data	Authors’ Comments
				WT/KI ^2^ KI/KI ^3^	WT/KI ^2^ KI/KI ^3^				
Kenna et al. [39]	AAV-delivered cloned human cDNA	n/a	Leu	n/a n/a	n/a n/a	Eyes expressing human D477G/RPE65 plus mouse WT RPE65 exhibit reduced ERG amplitudes but no structural defects compared to human WT RPE65 plus mouse WT RPE65.			
Li et al. [40]	CRISPR/Cas9 genome editing	Q475H c.1434T>C c.1435C>T	Met	50 3	285 465	As for WT under normal lighting.	As for WT under normal lighting; dark adaptation recovery of cones delayed after 6 months low light stress.	c.1430A>G in KI/KI ^3^ mice and in human minigene in vitro creates defective splicing.	Aberrant splicing of human RPE65 could greatly reduce isomerase levels or create a novel protein with toxic properties.
Shin et al. [53]	Homologous recombination	101bp FRT site insertion in intron 13	Met	53 22	160 n/a	Dark adaptation recovery of cones delayed; lipid droplet accumulation in RPE.	Very low quantity (≪25% WT) of RPE65 detected in RPE.	WT/KO ^4^ all-trans retinyl ester levels same as WT.	Comparison of WT/KI ^2^ and WT/KO ^4^ suggests dominant negative effect.
Kiang et al. (this review)	CRISPR/cas9 genome editing	n/a	Met	n/a n/a	n/a n/a	As for WT under normal lighting.	As for WT under normal lighting.	No effect of high cholesterol diet.	
Kiang et al. (this review)	Homologous recombination	Q475H	Leu	n/a n/a	n/a n/a	As for WT under normal lighting.	As for WT under normal lighting.	No delay in dark adaptation recovery was observed in either WT/KI ^3^ or KI/KI	

^1^ Approximate values taken from published figures; ^2^ heterozygous knock-in; ^3^ homozygous knock-in; ^4^ heterozygous knock-out.

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
