# Peer review of "Properties and Therapeutic Implications of an Enigmatic D477G RPE65 Variant Associated with Autosomal Dominant Retinitis Pigmentosa"

_genes, 2020, doi:10.3390/genes11121420_

Round 1

Reviewer 1 Report

This is an interesting review article devoted to the clinical and experimental studies related to the D477G variant of RPE65 that is uniquely associated with the autosomal dominant form of RP associated with RPE65.  The authors are well recognized experts in the field. 

The manuscript is generally well written.  Personally, I would have liked a better balance between the major sections, however.  More clinical images would be useful to show the phenotypic diversity, and it would be good to show a pedigree or two to better appreciate this autosomal dominant with reduced penetrance condition.  I am particularly interested in seeing the autofluorescence images of the individuals in Figure 1.  Patients with AR RPE65 mutations have almost no lipofuscin, what do patients with AD RPE65 show?  The animal studies section is overly long.  They lead with a paragraph on lines 194-198 on the weakness of animal models for retinal degeneration and then proceed with 6 pages of discussion of the various published models that generally proved to be unenlightening because of the very mild phenotype.  The final section on 9-cis-retinoid therapy is reasonably well done, but should include a discussion of the limitations of this approach, especially since this was a phase 1 study without placebo control. 

Other issues:

Line 48:  Rewrite more clearly and correctly.  11-cis-retinol is oxidized to 11-cis-retinal.

Line 60: docosahexaenoic acid is misspelled.

Line 142: "impaired" should be "impairment"

Line 420: "illicit" should be "elicit" 

Reviewer 2 Report

In this well-written and lucid review, the authors collate, present, summarise and draw novel insights from published and unpublished data relating to this unique dominantly acting pathogenic variant in RPE65. The review adds significantly to our understanding of mechanisms of this disease, with likely relevance to other inherited retinal diseases, including choroideremia (which can show a similar phenotype) and others. The authors also discuss future directions. The review in my view is an important addition to the literature.

The authors may wish to note that the fact that the phenotype is so different from recessive RPE65-associated disease (including the milder spectrum of that disease), and that carriers of heterozygous null mutations in RPE65 are normal, already suggests that the mechanism of disease is likely to be something other than a mere reduction in isomerase activity alone.

I have only very minor comments as follows:

Abstract, last sentence: “… those of choroideraemia”

“Those” here appears to refer to “clinical spectrum” which is singular. Perhaps consider omitting “those of”?

Also, “choroideremia” might be more true to the etymological origin of the word than “choroideraemia”. It does not share the same origin as the “aem” in haemoglobin and anaemia. However, both spellings are commonly used, so this does not necessarily need changing.

Line 59 – “isomerase lutein to meso-zeaxanthin,”. Should it be be “isomerise”?

Lines 106-107: “Indeed, it is likely that the observed increased electrophoretic mobility of D477G RPE65 compared to the WT protein and in silico predicted perturbations [31] reflect the amino acid substitution and that this changed nature of the protein may cause a dominant effect (discussed later).”

How could these features not be a reflection of the amino acid substitution? The phrase “it is likely” might be redundant here?

Line 131: I suggest adding a reference for the ISCEV standards, for example, Robson AG, Nilsson J, Li S, Jalali S, Fulton AB, Tormene AP, Holder GE, Brodie SE. ISCEV guide to visual electrodiagnostic procedures. Doc Ophthalmol. 2018 Feb;136(1):1-26. 

Line 141 “symptoms related to… is” should be “symptoms related to … are”

The discussion in Section 5.4 regarding other non-isomerase functions of RPE65 would not seem to explain why the other knock-in studies did not show much effect? Could the authors discuss/speculate why?

Also, the authors refer to possible abnormalities of cone opsin localisation. But this perhaps still does not explain why the peripheral retina is still more affected?

Last sentence (lines 525-526): “… by reducing photoreceptor damage caused by continued firing of non-liganded opsin apoproteins [58].”

The word “firing” is usually used in terms of action potentials down an axon or the discharge of a neuron at the synapse. A better word here might be “activity” or similar.
